# OpenReview forum: "Sound Probabilistic Safety Bounds for Large Language Models"
_ICLR.cc/2026/Conference — Submitted to ICLR 2026_

### Official Review · Reviewer_vuwU · 2025-10-25

**Soundness:** 4
**Presentation:** 3
**Contribution:** 2
**Rating:** 6
**Confidence:** 4

**Summary:**

The authors introduce a new framework for computing rigorous bounds on the probability of harmful outputs from a prompt to a large language model (LLM). They study classical Clopper-Pearson confidence intervals and propose an algorithm to prioritize harmful branches in the autoregressive generation procedure.

**Strengths:**

+ A pioneer work on studying probabilistic bounds for LLMs.
+ Good presentation even for readers outside the domain.

**Weaknesses:**

- Some parts are not clearly illustrated.
- The experiment setting is too simple, which may hinder the practical usage.
- The theoretical guarantee and proof are not given with the computing upper and lower bounds.

**Questions:**

1. Page 6, Line 306: Given that the generation tree of LLM is very large, the partial tree construction should not be that useful.

2. Page 7, Line 332: Do the "activated features" refer to all activation values inside the LLM? This may introduce high computational complexity.

3. Page 8: The experiment is only conducted in 1B-level models, the extendability is unclear. Moreover, the safety oracle is a finite set of words with a limited number of words.

4. A few related works should be discussed:
- Song, Da, Xuan Xie, Jiayang Song, Derui Zhu, Yuheng Huang, Felix Juefei-Xu, and Lei Ma. "Luna: A model-based universal analysis framework for large language models." IEEE Transactions on Software Engineering 50, no. 7 (2024): 1921-1948.
- Zhang, M., Goh, K. K., Zhang, P., Sun, J., Xin, R. L., & Zhang, H. (2024). LLMScan: Causal Scan for LLM Misbehavior Detection. arXiv preprint arXiv:2410.16638.

---

> ### Author Response · Authors · 2025-11-23
>
> Thank you for reading the paper!
>
> > Page 6, Line 306: Given that the generation tree of LLM is very large, the partial tree construction should not be that useful.
>
> As we explicitly acknowledged in the paper, our approach relies on constructing the generation tree. The constructed tree does not need to be exhaustive or exponentially large; our approach works even if a partial tree is available. To utilize this feature, we selectively construct only those branches of the tree that are (i) potentially harmful and (ii) have non-negligible probability mass. In practice, the vast majority of the tree either corresponds to harmless continuations or carries minuscule probability, and therefore does not need to be explored in order to compute a sound lower bound. This targeted expansion makes the approach tractable while preserving the soundness of the bound.
>
> > Page 7, Line 332: Do the "activated features" refer to all activation values inside the LLM? This may introduce high computational complexity.
>
> Similar to [1], we use the Residual Stream vectors that are of size 1024 or 2048 in the used LLMs. This does not introduce a computationally expensive overhead.
>
> [1] Arditi, Andy, et al. "Refusal in language models is mediated by a single direction." Advances in Neural Information Processing Systems 37 (2024): 136037-136083.
>
> > Page 8: The experiment is only conducted in 1B-level models, the extendability is unclear. Moreover, the safety oracle is a finite set of words with a limited number of words.
>
> We emphasize that the complexity of the safety oracle does not change the efficiency of our algorithm, as we use it in a black-box manner. We appreciate your suggestion and will include more experiments in the updated version of the paper.

---

### Official Review · Reviewer_ksh1 · 2025-10-28

**Soundness:** 1
**Presentation:** 1
**Contribution:** 1
**Rating:** 2
**Confidence:** 3

**Summary:**

This paper propose a framework to compute bounds on the probability that a given prompt to LLM generates harmful outputs. The authors begin by Clopper-Pearson confidence intervals and propose an algorithm that analyses features in latent space to prioritize the autoregressive generation tree.

**Strengths:**

1. **Good trial of math modelization of safety bounds**: The authors explore Clopper-Pearson exact confidence intervals and autoregressive generation tree to compute the bounds on the probability that a given prompt to LLM generates harmful outputs. This is a good trial to explore the theoretical basis behind LLMs safety.

**Weaknesses:**

1. The computed bound is not **rigorous** as claimed by the authors (line 11, 65), because they use approximate linearity features and computational budget to approximate the bound.
2. The experiment is very poor. It is more like a case study because only 2 cases are evaluated. Besides, the computed lower bound is still very low ($10^{-8} - 10^{-4}$). Considering the upper bound is at around $10^{-2}$, such a wide range cannot be used to interpret or improve LLMs safety.
3. What does $X_i$ of line 209 correspond to? Does it correspond to each generated token? If yes, why can it be assumed as i.i.d. Bernoulli random variable since each token depends on the preceding tokens?
4. In line 298, the authors assume that once a prefix of an output is harmful, every continuation extending this prefix is also harmful. But this is not always true because LLMs are observed to correct their wrong output prefix during inference [1]. Taking the same example in Figure 1, the output "You need to install, wait wait wait, this is illegal, I cannot assist you." has a harmful prefix but is harmless as a whole.
5. The presentation and organization of paper is poor: algorithm 1 is too long, experiment I is cut to different pages, and there is no conclusion section.

[1] Course-Correction: Safety Alignment Using Synthetic Preferences

**Questions:**

1. What does $X_i$ of line 209 correspond to? Does it correspond to each generated token? If yes, why can it be assumed as i.i.d. Bernoulli random variable since each token depends on the preceding tokens?

---

> ### Author Response · Authors · 2025-11-21
>
> Thank you for your comments.
>
> > 1. The computed bound is not rigorous as claimed by the authors (line 11, 65), because they use approximate linearity features and computational budget to approximate the bound.
>
> We respectfully disagree on this point. The resulting bound is mathematically rigorous and sound. Our framework guarantees a formal lower bound $p_L$ on the true harmful probability $p$: $p_L \le p$. The framework is sound because it only sums the probabilities of the explored and verified harmful subtrees. The use of heuristics and the use of a computationally-limited  budget only affects the tightness of the bound (i.e., how close $p_L$ is to $p$), not its formal correctness. We emphasize that the core contribution of our work is the establishment of a framework for rigorous probabilistic guarantees via a search process that is guided by efficient heuristics.
>
> > 2. The experiment is very poor. It is more like a case study because only 2 cases are evaluated. Besides, the computed lower bound is still very low ($10^{-8}-10^{-4}$). Considering the upper bound is at around $10^{-2}$, such a wide range cannot be used to interpret or improve LLMs safety.
>
> The current experiment serves as a proof-of-concept case study illustrating the framework's capability rather than a comprehensive safety evaluation. The primary takeaway is that even with a limited computational budget (only 2,000 generated tokens per experiment), our approach yields a non-trivial lower bound on the harmful probability (i.e., $p_L > 0$). While the bounds are currently not tight, they demonstrate the existence of harmful behaviours and provide a mathematically sound estimate of their minimum frequency.
> The tightness of the bounds can be improved by scaling up the computational resources (expanding the tree search). We will revise and update the paper by including the results with increased computational resources to demonstrate the framework's scalability and its ability to achieve tighter, more actionable bounds.
>
> > 3. What does $X_i$ of line 209 correspond to? Does it correspond to each generated token? If yes, why can it be assumed as i.i.d. Bernoulli random variable since each token depends on the preceding tokens?
>
> $X_i$ represents the full generated response (a sequence of tokens of length $L$) by the LLM in the $i$-th trial, given a fixed prompt. We are considering $n$ such trials, $X_1, X_2, \dots, X_n$. These samples are from the same large language model and prompt, and are samples independently from each other, hence they are i.i.d.
>
> > 4. In line 298, the authors assume that once a prefix of an output is harmful, every continuation extending this prefix is also harmful. But this is not always true because LLMs are observed to correct their wrong output prefix during inference [1]. Taking the same example in Figure 1, the output "You need to install, wait wait wait, this is illegal, I cannot assist you." has a harmful prefix but is harmless as a whole.
>
> We appreciate the insight and the reference to LLMs self-correction. In this work, we specifically focus on safety specifications where the first instance of harmful content constitutes a violation. For example, if the specification is "the LLM must never suggest a hacking tool," then the generation of the tool's name in the prefix (e.g., "You need to install scapy") is already considered a safety violation, irrespective of the subsequent self-correction ("wait wait wait, this is illegal..."). In the example, "You need to install, wait wait wait, this is illegal, I cannot assist you.", this will not be labeled as harmful by the oracle because there is no package mentioned in the response.
>
> > 5. The presentation and organization of paper is poor: algorithm 1 is too long, experiment I is cut to different pages, and there is no conclusion section.
>
> We will reformat the algorithm to be more concise and add a conclusion section in the final version of the paper. Thank you for your suggestion.

---

### Official Review · Reviewer_k5WE · 2025-10-29

**Soundness:** 2
**Presentation:** 3
**Contribution:** 3
**Rating:** 4
**Confidence:** 3

**Summary:**

This paper proposes a novel framework for computing rigorous bounds on the probability that a Large Language Model (LLM) generates harmful outputs when given a specific prompt. It first discusses the limitations of existing sampling-based methods, like Clopper–Pearson confidence intervals, which often yield trivial (zero) lower bounds because harmful events are rare in aligned LLMs. The core contribution is an algorithm that constructs a partial autoregressive generation tree, guided by a "harmfulness feature" vector computed in the latent space to prioritize the exploration of branches most likely to lead to harmful content. This approach efficiently computes non-trivial, mathematically sound lower bounds on the harmfulness probability, which is crucial for safety certification in high-stakes LLM applications. Experimental results demonstrate that this method consistently provides a superior lower bound compared to standard Monte Carlo and Clopper–Pearson techniques.

**Strengths:**

- The authors present a strong theoretical grounding of the work presented, while maintaining good flow and ease-of-read throughout the paper.
- The paper addresses relevant problem that has recently gained attention in the research community.

**Weaknesses:**

- There are some articles the authors might like to consider that introduce some related notions to the ones discussed in the paper. For example, in a more general sense, there is literature on domain certification (not necessarily safety certification, as the authors discuss, but could this be seen as a particular case of domain certification?):
	* Emde, C., Paren, A., Arvind, P., Kayser, M., Rainforth, T., Lukasiewicz, T., ... & Bibi, A. (2025). Shh, don't say that! Domain Certification in LLMs. arXiv preprint arXiv:2502.19320.
- I suggest taking a look at the related literature in the paper above, and clarifying how this work sets apart from other works such as:
	* Krause, B., Gotmare, A. D., McCann, B., Keskar, N. S., Joty, S., Socher, R., & Rajani, N. F. (2020). Gedi: Generative discriminator guided sequence generation. arXiv preprint arXiv:2009.06367.
	* Yang, K., & Klein, D. (2021). FUDGE: Controlled text generation with future discriminators. arXiv preprint arXiv:2104.05218.
	* Fonseca, J., Bell, A., & Stoyanovich, J. (2025). Safeguarding large language models in real-time with tunable safety-performance trade-offs. arXiv preprint arXiv:2501.02018.
- The authors claim they propose a method for computing "rigorous bounds" on the safety of an generated answer, given a certain prompt. However, their method relies on an oracle that is either manually defined with specific keywords, or a trained neural network classifier. While the former is not a scalable/generalizable approach, the latter is prone to its own limitations; jailbreak attempts could manipulate the scores produced by this neural network.
- On the same note, the experimental evaluations conducted seem insufficient. The authors merely present two examples using predefined keywords, while the neural network approach the authors mention previously is never actually reported. Furthermore, in the keywords list displayed, the existence of the word "scrapy" (experiment I), or the word "search" (experiment II), are not dangerous on their own, yet their existence in an output is sufficient for it to be considered dangerous according to the proposed method.
- In the two examples provided, it is impossible to accurately compare the varying top-k values since the temperature values are not the same.
- In addition, the provided repository appears to be empty (I see two files listed, a readme and a jupyter notebook, both yielding the error "The requested file is not found."). I cannot validate the reproducibility of this work, or validate any details I might be missing based on the actual implementation of the proposed method.

**Questions:**

- I'm not particularly convinced on the significance of the proposed method in a practical setting. Generally, deployed models use very low temperature (much lower than the reported values) and top-k settings for next-token prediction, which raises the question on whether the instances found would show up at all, at a non-near-zero probability. Could the authors provide an analysis or additional experimental results using significantly lower temperature settings (e.g., T << 0.4) than those reported in the experiments, or even T=0 (greedy decoding)? This would help demonstrate the utility of the method in configurations commonly used for stable deployment, where stochasticity is minimized.
- Given that the computed p_L​ is a mathematically rigorous lower bound on the true probability p, how does the framework's guarantee of rigor hold if the underlying safety oracle H itself is known to be imperfect or susceptible to manipulation, such as prompt injection or jailbreak attempts?

---

> ### Author Response · Authors · 2025-11-24
>
> Thank you for your comments and providing pointers to the papers. We emphasize that the contribution of our work is to quantify the safety of large language models, rather than to provide new safety techniques. We provide a rigorous measurement framework that enables evaluation and certification, distinct from inference-time safety methods.
>
> > On the same note, the experimental evaluations conducted seem insufficient. The authors merely present two examples using predefined keywords, while the neural network approach the authors mention previously is never actually reported. Furthermore, in the keywords list displayed, the existence of the word "scrapy" (experiment I), or the word "search" (experiment II), are not dangerous on their own, yet their existence in an output is sufficient for it to be considered dangerous according to the proposed method.
>
> We find responses that contain python packages harmful, even if they don't have direct harmful purposes (for instance, “scrapy”). The word “search” in the context often appears as “search for [some harmful approach/tool]”, but we agree that it is not dangerous on its own. We will update the safety oracle in the revised paper. Thank you for mentioning this.
>
> > In the two examples provided, it is impossible to accurately compare the varying top-k values since the temperature values are not the same.
>
> Thank you for your comment. We will include results with the same top-k and temperature values for comparability in the updated paper. Thank you for pointing this out.
>
> > In addition, the provided repository appears to be empty (I see two files listed, a readme and a jupyter notebook, both yielding the error "The requested file is not found."). I cannot validate the reproducibility of this work, or validate any details I might be missing based on the actual implementation of the proposed method.
>
> Sorry for your inconvenience. We checked the link again and it appears to work fine. As an alternative, please use the following link and hit the download button to get the file:
> https://sendgb.com/Ha7DGAIcbOq
>
> > I'm not particularly convinced on the significance of the proposed method in a practical setting. Generally, deployed models use very low temperature (much lower than the reported values) and top-k settings for next-token prediction, which raises the question on whether the instances found would show up at all, at a non-near-zero probability. Could the authors provide an analysis or additional experimental results using significantly lower temperature settings (e.g., T << 0.4) than those reported in the experiments, or even T=0 (greedy decoding)? This would help demonstrate the utility of the method in configurations commonly used for stable deployment, where stochasticity is minimized.
>
> We agree that some deployed systems rely on low-temperature generation settings. However,
>
> - Recent empirical studies (e.g., [1]) show that a broad range of LLM tasks maintain optimal or near-optimal performance at temperatures in the range ($0.5-1.0$). Our experiments were designed to evaluate model behavior under such widely used stochastic sampling settings.
>
> - At T=0, decoding becomes fully deterministic and the harmfulness probability collapses to a binary value: the model either always produces a harmful output (probability 1) or never does (probability 0). In this regime, there is no stochastic behavior to be evaluated by our framework.
>
> In the revised version, we will add experiments at reduced temperatures (T < 0.4), to assess how the lower-bound behavior scales as stochasticity decreases. Thank you for your comment.
>
> [1] Du, Weihua, Yiming Yang, and Sean Welleck. "Optimizing temperature for language models with multi-sample inference." arXiv preprint arXiv:2502.05234 (2025).

---

> > ### Author Response · Authors · 2025-11-24
> >
> > > Given that the computed p_L​ is a mathematically rigorous lower bound on the true probability p, how does the framework's guarantee of rigor hold if the underlying safety oracle H itself is known to be imperfect or susceptible to manipulation, such as prompt injection or jailbreak attempts?
> >
> > Our framework assumes access to a safety oracle $\mathcal{H}$. Relying on a safety oracle is consistent with prior work on quantifying LLM behavior (e.g., [2], [3]). The oracle in our work does not need to be universally robust: the oracle $\mathcal{H}$ only needs to be robust for the specific prompt under evaluation and therefore it allows for prompt-specific tools. Moreover, the oracle $\mathcal{H}$ in practice can even be realized by human annotation.
> >
> > Jailbreaking concerns typically arise in adversarial settings. In our framework, there is no external adversary - the prompt distribution, model, and oracle are controlled by the same party. Thus, robustness against adversarial prompt injection is not required for the validity of the bound in this measurement setting.
> >
> > We emphasize that even if $\mathcal{H}$ mislabels some harmful responses as harmless, the computed $p_L$ ​remains a valid lower bound on the true harmfulness probability $p$. In such cases, $p_L$ ​will underestimate $p$, but it will never exceed it, and therefore still preserves the rigor of a lower-bound certificate.
> >
> > We will elaborate on these aspects and their implications in the revised version of our paper.
> >
> > [2] Wu, Gabriel, and Jacob Hilton. "Estimating the probabilities of rare outputs in language models." arXiv preprint arXiv:2410.13211 (2024).
> >
> > [3] Jones, Erik, et al. "Forecasting rare language model behaviors." arXiv preprint arXiv:2502.16797 (2025).

---

### Official Review · Reviewer_jNJK · 2025-11-01

**Soundness:** 1
**Presentation:** 1
**Contribution:** 2
**Rating:** 4
**Confidence:** 4

**Summary:**

This paper introduces framework to deduce non trivial bounds for LLMs to samples harmful (as defined an oracle) responses to set of prompts. The proposed framework poses autoregessive sampling from a language model as a tree structure with each node being a token and subsequent edge leading to next token thereby making each unique path to a leaf being a unique sequence. Each sequence probability is the product of prob along one path of the tree (albeit very large tree due to vocab size). This tree depicts the joint distribution of sequences from given prompt. The framework depends on 3 "observations" (I would call them maybe axioms): 1) if a equence becomes labelled as harmful at one point in the sequence it can never go back to being "unharmful", 2) "the sum of probabilities of harmful leaves in this subtree provides a lower bound on the true harmfulness probability p" and 3)

**Strengths:**

- I really liked the reading this paper, problem formulation for clear and concise, I also appreciate the covering of PAC bounds.
- I think the proposed framework is intuitive in the sense it beam searches the most harmful responses and increases the lower bound of unsafe responses based on that.
- This method translates really well to empirical experiments and thus is more relevant.

**Weaknesses:**

See questions

**Questions:**

- Is the assumption that there is an oracle that labels harmful from non-harmful completion exists too strong?
- How can this lower bound be leveraged? Like in my first point, one needs an oracle to for this, and if you have that then you just piecewise update the sampler to never sample generations which are harmful (i.e. H(prompt)= 1). And then your upperbound drops to 0 i.e. you never sample harmful response for given oracle function H. I'm not sure what is the way to leverage this framework at frontier scale? Is this just a mental model to think about bounds? I like this but I'm not sure what exact value this work provides.
- I would like the second limitations addressed within this version of the paper for it to be accepted. For the reader, current experiments computes bounds on per prompt basis, but we need to see it from a distribution of prompts and perhaps see a histogram of different prompt distribution to make better sense. Currently, the experiments are too empty.
- The paper reaches word limit without discussing much results. I think paper writing needs surgery to better convery + experimentally convince the reader.

---

> ### Author Response · Authors · 2025-11-21
>
> Thank you for your time to read the paper and thank you for your comments!
>
> > Is the assumption that there is an oracle that labels harmful from non-harmful completion exists too strong?
>
> We assume the existence of a black-box oracle capable of classifying model responses as harmful or non-harmful. This oracle is implementation-agnostic, practically realized by an LLM classifier, a heuristic filter, or even human labeling. This assumption is standard and necessary for this line of research, as evidenced by its use in closely related works [1] and [2].
>
> [1] Wu, Gabriel, and Jacob Hilton. "Estimating the probabilities of rare outputs in language models." arXiv preprint arXiv:2410.13211 (2024).
>
> [2] Jones, Erik, et al. "Forecasting rare language model behaviors." arXiv preprint arXiv:2502.16797 (2025).
>
>
> > How can this lower bound be leveraged? Like in my first point, one needs an oracle to for this, and if you have that then you just piecewise update the sampler to never sample generations which are harmful (i.e. H(prompt)= 1). And then your upperbound drops to 0 i.e. you never sample harmful response for given oracle function H. I'm not sure what is the way to leverage this framework at frontier scale? Is this just a mental model to think about bounds? I like this but I'm not sure what exact value this work provides.
>
> Thank you for raising this point. Our intention is for the proposed framework to serve as a rigorous methodology for measuring the harmfulness of large language models, rather than as a mechanism for inference-time safeguarding. For example, a safety evaluation team can certify that a model’s harmfulness probability does not exceed a required threshold by using an appropriately defined oracle $\mathcal{H}$—which may be implemented via prompt-specific rules, or human labeling. While we agree that the method is not directly applicable to runtime mitigation, we believe it provides a principled and quantitative foundation for model-level evaluation and certification.
> > I would like the second limitations addressed within this version of the paper for it to be accepted. For the reader, current experiments computes bounds on per prompt basis, but we need to see it from a distribution of prompts and perhaps see a histogram of different prompt distribution to make better sense. Currently, the experiments are too empty.
>
> We appreciate your suggestions and will include more experiments in the paper in the updated version.
>
> > The paper reaches word limit without discussing much results. I think paper writing needs surgery to better convery + experimentally convince the reader.
>
> We appreciate your suggestion and will reformat the paper in the updated version.

---

### Meta-Review · Area_Chair_TQE5 · 2026-01-06

**Summary:**

This paper proposes a framework for computing rigorous lower bounds on the probability of harmful outputs from LLMs. While the idea has novelty, it suffers from fundamental flaws and does not meet the acceptance threshold.

The main issues are as follows:
1）Limited Practical Utility: The framework relies on a strong assumption of an "oracle" to judge harmful content. If a perfect oracle already exists, safety measures could be implemented directly, casting doubt on the practical value of this lower bound. The authors argue it is intended for offline evaluation, but this use case is not clearly justified in the paper.

2） Severely Inadequate Experiments: The experiments are based on only two simple case studies and a crude keyword-based oracle, failing to demonstrate the method's effectiveness in real-world scenarios. The obtained probability lower bounds are too wide, offering little practical guidance.

3）Core Assumptions May Be Invalid: A reviewer pointed out that models may self-correct harmful prefixes, which challenges the key assumption that "a harmful prefix inevitably leads to a harmful continuation." This undermines the theoretical rigor of the entire lower-bound framework.

In summary, although the authors have indicated they will revise the experiments and text, the current version has significant shortcomings in conceptual contribution, empirical support, and theoretical soundness.  The authors are advised to thoroughly refine the theoretical assumptions and conduct sufficient experimental validation before resubmitting.

**Reviewer Concerns:**

Concerns Addressed：
1）Justification of Assumptions: The authors cited relevant work in the field to argue for the reasonableness and necessity of the oracle assumption in this research.
2） Clarification of Method's Purpose: It was clarified that the framework aims to provide a quantitative tool for model safety evaluation and certification, not for inference-time guarding, addressing questions about its application scenario.
3）Technical Queries: Effective explanations or solutions were provided for specific technical issues such as code accessibility, feasibility of partial tree construction, and computational cost of features.


Outstanding Concerns：

1）All reviewers noted that the current experiments are extremely weak (only two cases, keyword-based oracle, no coverage of prompt distributions, lacking low-temperature/real-world deployment scenario tests). The authors only promised "to conduct more experiments" in the future, without providing any new data, analysis, or results, leaving this core weakness completely unaddressed in this review cycle.
2）Although the purpose was clarified, how this rigorous lower bound provides unique, actionable evaluation value given the pre-existence of an oracle still lacks a convincing explanation.
3）Regarding the core assumption that "a harmful prefix makes the entire path harmful," the authors defended it based on specific safety specifications. However, they failed to eliminate the reviewers' fundamental doubts about its general validity, which undermines the theoretical foundation of the framework.
4） The scalability of the method to larger models and its comparative analysis with a broader range of related work have not been validated or discussed.

Although the authors responded to many specific queries, they failed to provide substantial progress to address the reviewers' fundamental doubts regarding the most critical aspects: empirical support, theoretical persuasiveness, and practical value.

**Reviewer Scores:**

Reviewer jNJK (Score: 4 → 3)
The rebuttal addressed some conceptual points but failed to provide new data for the critically weak experiments. Given the emphasis on empirical validation, the reviewer would likely move from borderline to clear reject.

Reviewer k5WE (Score: 4 → 3)
While code access was fixed and oracle reliability was argued, the core experimental flaws (lack of cases, low-temperature tests) remained unproven. The reviewer would likely shift toward reject based on insufficient evidence.

Reviewer ksh1 (Score: 2 → 2)
Fundamental disagreements on theoretical soundness and the harmful prefix assumption were not resolved. No new experimental evidence was offered, reinforcing the original reject stance.

Reviewer vuwU (Score: 6 → Likely 4 or below)
Technical clarifications were given, but key concerns about simplistic experiments and scalability were only met with vague promises. The reviewer would likely lower the score to borderline or reject.


A full discussion would strengthen the rejection consensus. The rebuttal provided promises and theoretical defenses but no substantive evidence to address core criticisms, leading borderline reviewers to lean toward reject.

---

### Decision · Program_Chairs · 2026-01-26

Reject